# Utilizing Mask R-CNN for Solid-Volume Food Instance Segmentation and Calorie Estimation

**Yanyan Dai, Subin Park**  **and Kidong Lee ***

Robotics Department, Yeungnam University, Gyeongsan 38541, Korea
* Correspondence: kdrhee@yu.ac.kr; Tel.: +82-10-2418-2635

**Abstract:** To prevent or deal with chronic diseases, using a smart device, automatically classifying food categories, estimating food volume and nutrients, and recording dietary intake are considered challenges. In this work, a novel real-time vision-based method for solid-volume food instance segmentation and calorie estimation is utilized, based on Mask R-CNN. In order to address the proposed method in real life, distinguishing it from other methods which use 3D LiDARs or RGB-D cameras, this work applies RGB images to train the model and uses a simple monocular camera to test the result. Gimbap is selected as an example of solid-volume food to show the utilization of the proposed method. Firstly, in order to improve detection accuracy, the proposed labeling approach for the Gimbap image datasets is introduced, based on the posture of Gimbap in plates. Secondly, an optimized model to detect Gimbap is created by fine-tuning Mask R-CNN architecture. After training, the model reaches AP (0.5 IoU) of 88.13% for Gimbap1 and AP (0.5 IoU) of 82.72% for Gimbap2. mAP (0.5 IoU) of 85.43% is achieved. Thirdly, a novel calorie estimation approach is proposed, combining the calibration result and the Gimbap instance segmentation result. In the fourth section, it is also shown how to extend the calorie estimation approach to be used in any solid-volume food, such as pizza, cake, burger, fried shrimp, oranges, and donuts. Compared with other food calorie estimation methods based on Faster R-CNN, the proposed method uses mask information and considers unseen food. Therefore, the method in this paper outperforms the accuracy of food segmentation and calorie estimation. The effectiveness of the proposed approaches is proven.

**Keywords:** food instance segmentation; solid-volume food calorie estimation; convolutional neural network; mask R-CNN; deep learning



## 1. Introduction

Despite advances in medicine in recent years, the number of people affected by chronic diseases remains high, due to their irregular eating habits and unhealthy lifestyles. Some of the prevalent chronic diseases include obesity, hypertension, hyperlipidemia, cardiovascular diseases, blood sugar, and different kinds of cancers [1]. To prevent or deal with chronic diseases, people usually record their dietary intake to estimate nutrient consumption. Many dietary mobile applications require users to manually enter their dietary intake. Using a smartphone camera, automatically classifying food categories, and estimating their volume and nutrients is still considered a challenge. This paper proposes a novel real-time vision-based method for solid-volume food instance segmentation and calorie estimation, using a simple smart camera.

Food recognition is a very complex task and has the following challenges. The non-rigid structure of foods and variations within the source complicates the correct classification of foods [1]. Since different ingredients may look very similar, it causes food recognition problems. Or due to different cooking methods, the same ingredient is recognized as a different food. Therefore, food recognition has become an attractive research topic in computer vision [2–4]. Compared with food recognition, calorie estimation of food is known as more a complex task. To estimate calories, it is necessary to correctly recognize

the food. In addition, even if the food is recognized correctly, it is difficult to estimate the volume of the food, especially if there is occlusion among the food. Some food does not have regular volumes, making it harder to estimate calories. In addition, depending on different types of cameras and lighting conditions, image quality and calibration methods affect the estimation results. Previous work in [5] talks about the need to translate, rotate, and scale the pre-constructed 3D food models to make the contour of the templates match the food. This method achieved an average of 79.05% accuracy in volume estimation. In [6], a method is proposed based on 3D reconstruction and a deep learning algorithm to consume food items. In the experiment, a depth camera records the rotating food items to get the ground truth volume of food. In [7], the network based on Resnet-50 is trained using RGB-D images as the input. Refs. [5–7] all require RGB-D cameras for food volume prediction, which are not portable and difficult to apply to daily life. In order to address the proposed method in real life, this paper only needs to use a simple monocular camera to take pictures and combines it with the calibration method to predict food calories. In this paper, Gimbap is selected as an example of a solid-volume food for discussing food recognition and calorie estimation. Gimbap is a Korean dish made of cooked rice with ingredients such as vegetables, fish, and meat, rolled into bite-sized slices in a Gim [8]. To make Gimbap, grilled Gim is placed on a bamboo Gimbap roller with a thin layer of rice on top. Then, some ingredients are placed on the rice and it is rolled into a cylindrical shape. Typically, the length of the cylinder is around 19 cm, and the diameter is around 4 cm. Based on research in this paper, the calories of the other shaped food can be evaluated.

Computer vision in machine learning has become a popular research topic recently [9–11]. Computer vision contains many research tasks: image classification, object detection, semantic segmentation, and instance segmentation. Object classification requires binary labels indicating whether objects are present in an image [12]. Deep Convolutional Neural Networks, such as AlexNet [13], ResNet [14], and EfficientNet [15], are used to extract features. For object detection, the object's specified class and its localization in the image are obtained. A bounding box presents an object's location. Based on [16], the object detection model includes two types: a one-stage method and a two-stage method. One-stage models consist of YOLO [17], Efficient Det [18], and CenterNet [19]. Two-stage models include R-CNN [20], Fast R-CNN [21], Faster R-CNN [22], and the Feature Pyramid Network (FPN) [23]. For semantic segmentation, each pixel of an image should be labeled as belonging to a category, such as dog, cat, and sheep. The same object does not need to be segmented separately. Instance segmentation is a combination of object detection and semantic segmentation. Mask R-CNN [24] is the most commonly used instance segmentation algorithm. Mask R-CNN performs pixel-level segmentation by adding a branch to Faster R-CNN that outputs a binary mask indicating whether a given pixel is part of the target object. The branch is a fully convolutional network based on convolutional neural network feature maps. There is a growing interest in the research of Mask R-CNN as well as various applications. In [16], strawberry diseases are detected with low cost and good accuracy. In [25], in order to count the plants and calculate the size of plants, the image taken by a drone is processed by Mask R-CNN. In [26], using Mask R-CNN, the waterline is detected and analyzed in the sports area. In [27], it generates a synthetic dataset for scale-invariant instance segmentation of food materials using Mask R-CNN. In [28], Mask R-CNN with data augmentation is used for food detection and recognition. However, both Refs. [27,28] do not consider the calorie estimation problem. In [29], a bottom-up regime is used to learn category-level human semantic segmentation and to estimate the multi-person pose.

TensorFlow's Object Detection API is a tool for building, training, and deploying object detection models. In most instances, training an entire convolutional network from scratch requires large datasets and takes time. This problem can be solved by using the advantage of transfer learning with a pre-trained model using the TensorFlow API. This work used TensorFlow's object detection API for Gimbap recognition. Mask R-CNN model

returns both the bounding box and the mask for each detected Gimbap. Based on the mask result, since only Gimbap area is detected, the calorie estimation can be more correct.

The main contributions of this paper: 1. Based on the posture of Gimbap in plates, we created the labeling approach for the Gimbap image datasets to process the Gimbap food instance segmentation and calorie estimation system. After annotation, the statistics of the dataset are concluded. 2. An optimized model is created by fine-tuning Mask R-CNN architecture for efficient instance segmentation of Gimbap Korean Food. In order to address the proposed method in real life, distinguishing it from other methods which use 3D LiDARs or RGB-D cameras, this work applies RGB images to train the model and uses a simple monocular camera to test the result. 3. Combining instance segmentation results and calibration results, a calorie estimation approach is approached for solid-volume food. The calorie estimation approach can be extended to any solid-volume food, such as pizza, cake, burger, fried shrimp, oranges, and donuts. 4. The experiment shows the effectiveness of the proposed approaches. After training, the model reaches AP (0.5 IoU) of 88.13% for Gimbap1 and AP (0.5 IoU) of 82.72% for Gimbap2. mAP (0.5 IoU) of 85.43% is achieved. The Gimbap's calories are correctly evaluated. Compared with other food calorie estimation methods based on Faster R-CNN, the proposed method uses the mask result from the Mask R-CNN algorithm and considers unseen food. The comparison result shows the outperforms in the accuracy of food segmentation and calorie estimation.

The remaining article is written below. Section 2 outlines instance segmentation and calorie estimation system. Section 3 represents the Mask RCNN algorithm. Section 4 proposed the calorie estimation approach. Section 5 shows the experimental results.

## 2. Instance Segmentation and Calorie Estimation System Overview

As shown in Figure 1, the Instance Segmentation and Calorie Estimation System includes the following steps: (1) Image acquisition and processing using a simple monocular camera; (2) Instance segmentation using the Mask R-CNN algorithm; (3) Calorie estimation.

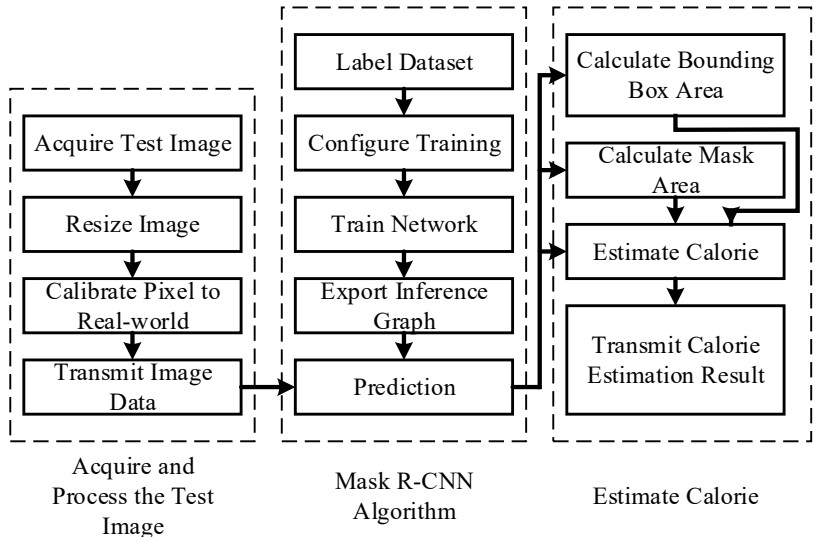

**Figure 1.** Flowchart of Gimbap Instance Segmentation and Calorie Estimation System.

In step (1), the image is resized to $640 \times 480 \times 3$, in order to increase transmission speed. In the calibration pixel to real-world block, the contour of the coin is detected. The contour's minimum and maximum pixels on the *x*-axis $(min_x, max_x)$ and on the *y*-axis $(min_y, max_y)$ are calculated, separately. Based on the diameter of the coin, the physical sizes of each pixel on the *x* and *y* axes of the image are calculated as (1) and (2), where $L_{ppx}$ and $L_{ppy}$ are the physical sizes corresponding to each pixel on the *x* and *y* axes. $D_c$ is the diameter of the coin.

$$L_{ppx} = D_c / (max_x - min_x) \tag{1}$$

$$L_{ppy} = D_c / \left( max_y - min_y \right) \tag{2}$$

The calibration result is shown in Figure 2. The cropping image is used as a calibration area, to reduce the calculation time. The big green area is the calibration area. The calibration tool, such as a coin, should be in this area. The blue line presents the contour of the coin. Based on this contour, the physical sizes of each pixel on the *x* and *y* axes are calculated.

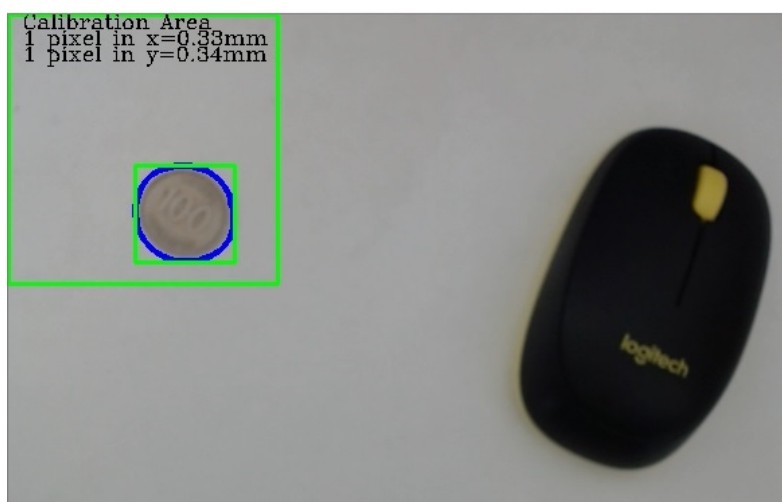

**Figure 2.** Calculate the physical size corresponding to each pixel on the *x* and *y* axes. The big green area is the calibration area. The blue line presents the contour of the object inside the small green box.

For image data transmission, the network topology user datagram protocol (UDP) is used and set up. Message Queuing Telemetry Transport and Data Distribution Services provide real-time behavior, so they are considered to be used.

### 3. Mask R-CNN Algorithm

*3.1. Prepare Datasets*

The Korea Food Image database, provided by Korea Institute of Science and Technology (KIST), is used for training and testing. A total of 800 Gimbap images are used as training images, the other 200 images are testing images. Labelme software is used to label the edge contours of Gimbap in images with label points. Different labels are used for classification. All the standard information of each image, for example, label name and edge points' coordinates, is saved to a json file corresponding to the original image.

Regarding classification, if there are many classes but the features of classes are similar, it will cause an issue that one thing is recognized as different classes, simultaneously. Based on the posture of the Gimbap in plates, there are two kinds of labels. As shown in Figure 3a, if the Gimbap is cut into pieces, the vegetables and rice are shown in the image, and it will be labeled as Gimbap1. The surface area $s_{ap1}$ of Gimbap1 is in range of [12.56, 28.26] cm$^2$. We define $s_{ap} = 19.625$ cm$^2$ in experimental section. As shown in Figure 3b, if the Gimbap is cut into pieces and placed obliquely, it will be labeled as Gimbap1. The surface area $s_{ar1}$ is around 125 cm$^2$. If the roll of Gimbap is not cut as shown in Figure 3c, and seaweed is shown in the image, it will be labeled as Gimbap2. If the roll of Gimbap is cut but only seaweed is shown in Figure 3a,b, it will be labeled as Gimbap2. For Gimbap 2, the surface area $s_{ar2}$ is around 95 cm$^2$. One roll of Gimbap is a unit for labelling. Or one piece of Gimbap is a unit.

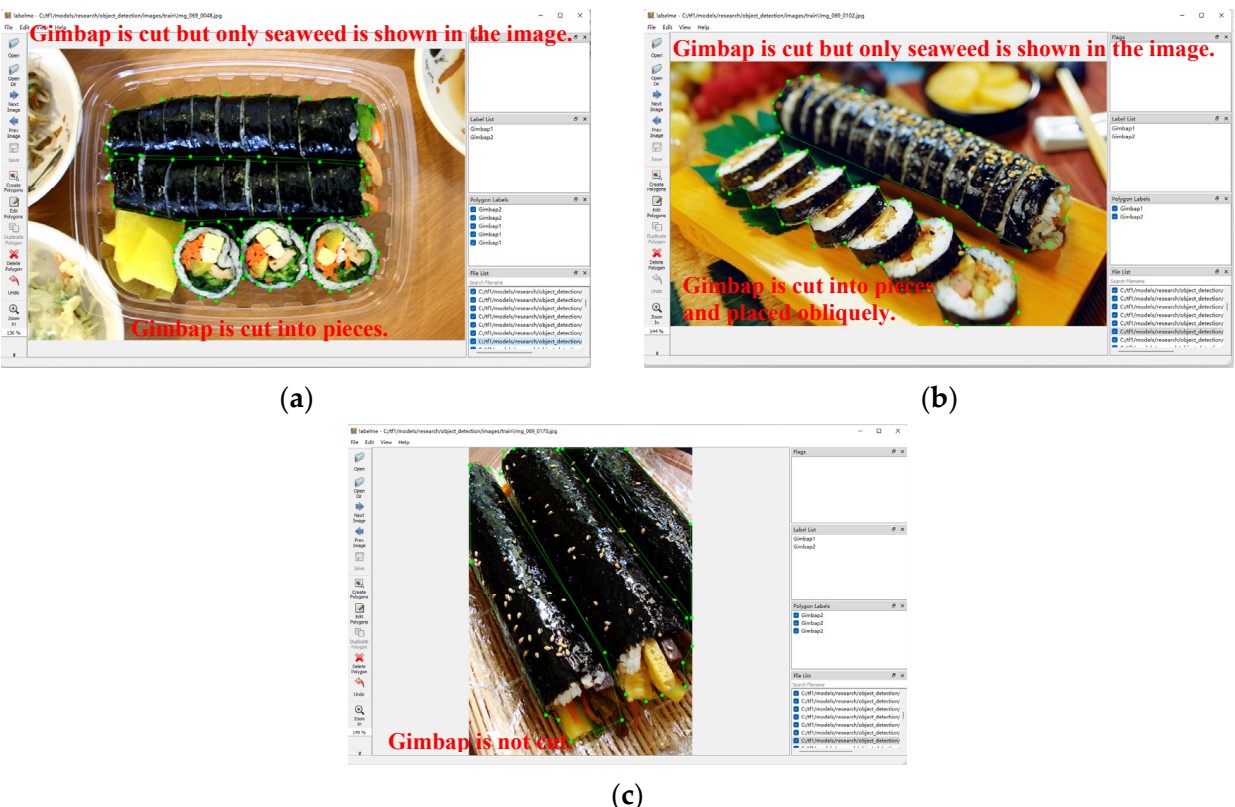

**Figure 3.** Classification of Gimbap. (a) The Gimbap is cut into pieces, the vegetables and rice are shown in the image, and it will be labeled as Gimbap1; (b) the Gimbap is cut into pieces and placed obliquely, it will be labeled as Gimbap1; (c) the roll of Gimbap is not cut and seaweed is shown in the image, it will be labeled as Gimbap2.

As shown in Figure 4a, labelme is used to open the original image and the corresponding json file. The image is divided into two parts by labeling points: the inside of the labeling points are different kinds of Gimbap, and the others are background. As shown in Figure 4b, different Gimbaps in the image will be covered with different color masks. All json files of the labeled images are combined into a json file, containing the labeling information of all labeled images, then they are converted into COCO datasets and inputted into the network for training.

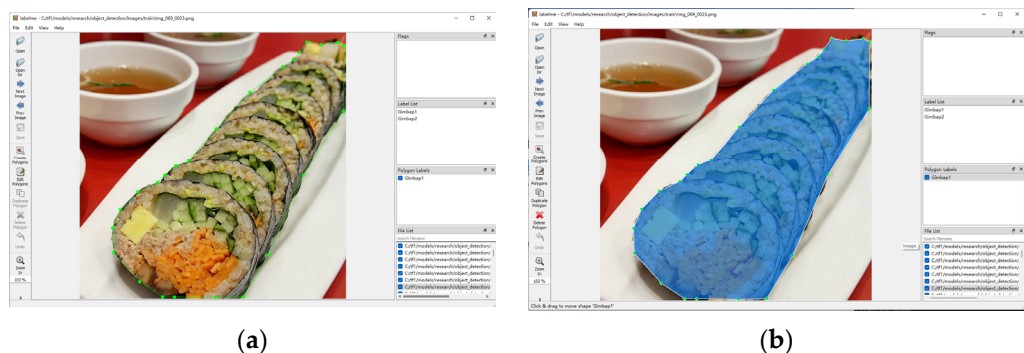

**Figure 4.** Label image. (**a**) Schematic Diagrams of the Gimbap Label. (**b**) Labeled Gimbap Sample.

After the annotation process, the statistics of the dataset are shown in Table 1. Among the 800 training images, there are 1318 Gimbaps labeled as Gimbap1, and 655 Gimbap2 labeled as Gimbap2. There are 278 negative food items, which are food but do not belong to

Gimbap. The test dataset contains 200 images. It includes 385 Gimbaps labeled as Gimbap1, 106 Gimbaps labeled as Gimbap2, and 56 negative food items.

**Table 1.** Dataset statistics after the annotation process.

| Collection | Number of Gimbaps Labeled as Gimbap1 | Number of Gimbaps Labeled as Gimbap2 | Number of Other Foods |
|---|---|---|---|
| **Train** | 1318 | 655 | 278 |
| **Test** | 385 | 106 | 56 |

### 3.2. Mask R-CNN Architecture

The Mask R-CNN can be divided into four main structures: backbone, Region Proposal Networks (RPN), region of interest (ROI) classification and bounding-box regression, and segmentation mask, as shown in Figure 5. The backbone structure of Mask R-CNN uses one or two commonly used convolutional neural networks to extract features from training images. Different deep learning networks have different feature extraction effects for different objects. When training with the same data, using Inception is faster, but using Resnet101 is more accurate than other networks such as Resnet50. In this paper, due to speed consideration, Inception_v2 is used and the result is acceptable.

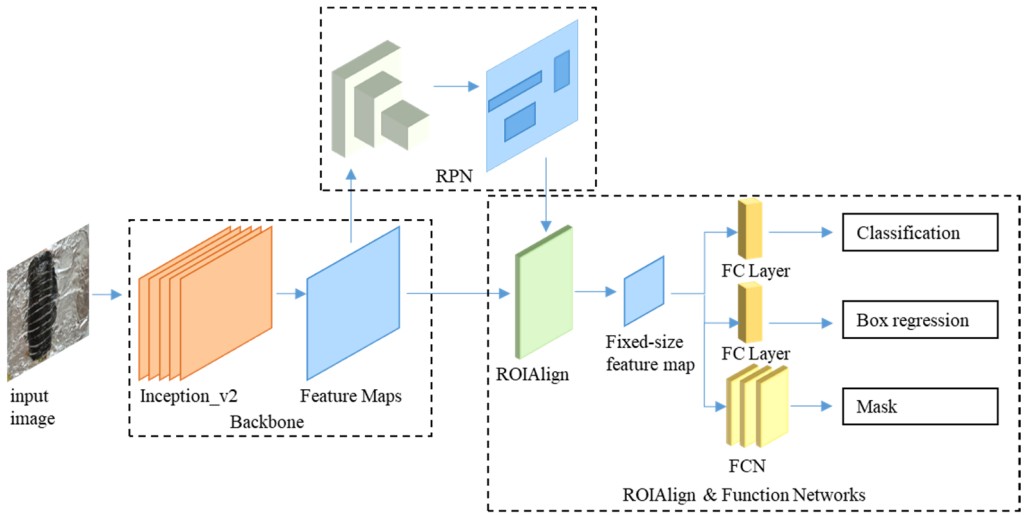

**Figure 5.** Architecture of Mask R-CNN.

The Inception_v2 module is designed to reduce the complexity of the convolution network. The Inception_v2 architecture consists of three main features: (1) it introduces two additional auxiliary classifiers in the middle of the network to address the vanishing gradient problem; (2) it has a deeper wider network architecture; (3) in order to solve the information loss problem caused by reducing the input size, the network upgrades Inception_v1 by using two $3 \times 3$ convolutions instead of one $5 \times 5$ convolutions. The convolution performance is improved by performing $3 \times 3$ convolutions [30].

RPN is a typical binary classification network. Its role is to classify images into two categories: target Gimbap and background. The Gimbap with boxes, which fit the size of the Gimbap as much as possible is framed. At this time, only the approximate area containing the target Gimbap and background can be distinguished. It is impossible to classify and segment the target Gimbap in detail. After RPN, one or more regions containing the target Gimbap can be obtained. Regions are input to ROIAlign and pooled into a fixed-size feature map.

### 3.3. Mask R-CNN Lost Function

ROIAlign classifies and locates each ROI. It uses a bilinear interpolation method in the feature map mapping process to precisely match the spatial corresponding of each pixel, so that each ROI is converted into a fixed-size feature map. Fixed-size feature maps are input into two branches. One of the branch networks performs target Gimbap recognition through an ROI classifier and bounding box regression. Both the classifier and the bounding box regression consist of a fully connected layer. A fully connected layer (FC layer) is used as an ROI classifier to classify the specific Gimbap categories, and the other fully connected layer acts as a bounding box regression to adjust the center point position and aspect ratio of the ROI, in order to detect the target Gimbap more accurately. The other branch network is composed of a fully convolutional network (FCN) to generate a segmentation network. The network will generate a mask with the same size and shape as the target Gimbap to segment the target Gimbap image. Finally, an image is obtained, containing the target Gimbap category and target Gimbap segmentation mask.

$$L = L_c + L_l + L_m \tag{3}$$

where $L$ is a total loss of the network; $L_c$ is classification loss, which is used to measure the accuracy of network classification; $L_l$ is localization loss, which is used to measure the frame positioning accuracy; and $L_m$ is mask loss, which is used to measure the accuracy of the mask position.

For each detection class $u$, the logarithm of the SoftMax loss function is used to calculate $L_c$.

$$L_c(p, u) = -log_2(p_u) \tag{4}$$

where $p = (p_0, p_1, \ldots, p_k)$ is the result of the Softmax function.

$L_l$ is calculated based on $smooth_{L_l}$ loss function.

$$L_l(t^u, v) = \sum_{i \in \{x,y,w,h\}} smooth_{L_l}(t_i^u - v_i) \tag{5}$$

where $v = (v_x, v_y, v_w, v_h)$ is the coordinates of the real bounding box of the target to be detected. $t^u = \left(t_x^u, t_y^u, t_w^u, t_h^u\right)$ the bounding box coordinate correction for the target of class $u$.

The $smooth_{L_l}$ function is defined as (6).

$$smooth_{L_l}(x) = \begin{cases} 0.5x^2 \ if |x| < 1 \\ |x| - 0.5 \ otherwise \end{cases} \tag{6}$$

The output dimension of the Mask branch for each ROI is K m$^2$, where K is the number of class and $m \times m$ is object binary mask size. A per-pixel sigmoid is applied, and $L_m$ is the average binary cross-entropy loss. For the ROI with class K, $L_m$ is computed only on the Kth Mask, and other Mask outputs are not counted in the loss.

## 4. Calorie Estimation Approach

The most common vegetable Gimbap contains 350–400 kcal. If different ingredients are used, such as tuna, cheese, and tempura, the calorie is close to 500 kcal. In this research, the calorie of one roll of basic Gimbap is defined as $cal_r = 375$ kcal. Usually, one roll of Gimbap is sliced into 10–12 pieces. Therefore, the calorie of one piece of Gimbap is defined as $cal_p = 35$ kcal. A Gimbap calorie estimation approach is proposed in Figure 6.

One variable of the detection result is boxes $[y_{min}, x_{min}, y_{max}, x_{max}]$, which is the bounding box coordinates. The bounding box area is calculated as (7)–(9):

$$bbox_x = (x_{max} - x_{min})Img_w \tag{7}$$

$$bbox_y = (y_{max} - y_{min})Img_h \tag{8}$$

$$bbox_{area} = bbox_x \times bbox_y \tag{9}$$

where $Img_w$ and $Img_h$ are the width and height of the test image.

Based on the output binary mask matrix, the binary image is created. Pixels with values more than 0 are extracted. Using OpenCV findContours function, the mask contour is obtained. Then mask contour area $cnt_{area}$ is calculated, based on OpenCV contourArea function. The area ratio is computed as:

$$area_{ratio} = \frac{cnt_{area}}{bbox_{area}} \tag{10}$$

As discussed in Section 2, the calibration is processed by calculating the physical size of each pixel on the $x$ and $y$ axes. For calculating Gimbap calories block, it contains several cases, based on the class of Gimbap. First, a gain $gb$ is calculated as follows:

$$gb = \begin{cases} \frac{bbox_x / bbox_y}{area_{ratio}} & if\ bbox_x > bbox_y \\ \frac{bbox_y / bbox_x}{area_{ratio}} & if\ bbox_y > bbox_x \end{cases} \tag{11}$$

The pseudo-code to calculate Gimbap calories is as Figure 7, based on (11). The Gimbap class is obtained from the Mask R_CNN classification result. During the annotation process, class 1 is labeled as Gimbap1, and class 2 is labeled as Gimbap2. The Total_Calorie_of_Gimbap is equal to zero. It is accumulated based on the number of detections in one image. Finally, the calorie estimation of the combination of Gimbap is represented by Total_Calorie_of_Gimbap.

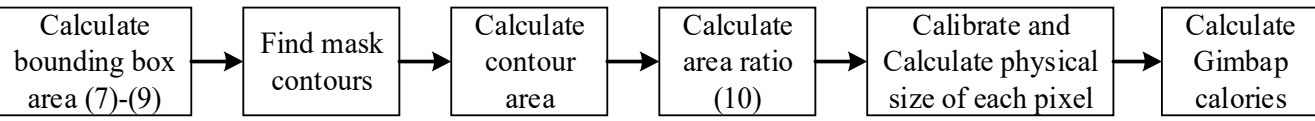

**Figure 6.** Gimbap calorie estimation diagram.

The calorie estimation approach can be extended in any solid-volume food, such as pizza, cake, burger, fried shrimp, oranges, and donuts. Based on the Mask R-CNN algorithm, the mask area $cnt_{area}$ and bounding box coordinates $[y_{min}, x_{min}, y_{max}, x_{max}]$ are obtained. Depending on the calibration process, physical sizes corresponding to each pixel on the $x$ and $y$ axes $L_{ppx}$ and $L_{ppy}$ are obtained. The solid-volume food's surface area can be calculated within a certain range. If the calorie of each piece of food is defined, then the calorie of the food in the image can be estimated.

```
Total_Calorie_of_Gimbap = 0
## Gimbap is cut into pieces.
## The mask of the Gimbap is round.
if class=1 and gb<1.5:
```

$$calorie = cal_p \times \frac{cnt_{area}}{\frac{s_{ap1}}{L_{ppx} \times L_{ppy}} \times s_g}$$

```
        Total_Calorie_of_Gimbap = Total_Calorie_of_Gimbap + calorie
## Gimbap is cut into pieces and placed obliquely.
## The mask of the Gimbap is rectangle.
else if class=1 and gb≥1.5:
```

$$calorie = cal_r \times \frac{cnt_{area}}{\frac{s_{ar1}}{L_{ppx} \times L_{ppy}} \times s_g}$$

```
        Total_Calorie_of_Gimbap = Total_Calorie_of_Gimbap + calorie
## Gimbap is not cut. Or Gimbap is cut but only seaweed is shown in the image.
## The mask of the Gimbap is rectangle.
else if class=2:
```

$$calorie = cal_r \times \frac{cnt_{area}}{\frac{s_{ar2}}{L_{ppx} \times L_{ppy}} \times s_g}$$

```
        Total_Calorie_of_Gimbap = Total_Calorie_of_Gimbap + calorie
## The total calorie is obtained from the value of Total_Calorie_of_Gimbap
```

**Figure 7.** The pseudo-code to estimate Gimbap calorie, where $s_{ap1}$, $s_{ar1}$ and $s_{ar2}$ are the surface area of the labeled Gimbap. $L_{ppx}$ and $L_{ppy}$ are the physical sizes corresponding to each pixel on the *x* and *y* axes, based on calibration process. $s_g$ is a constant scale gain.

## 5. Experimental Results

*5.1. Gimbap Instance Segmentation Using Mask R-CNN*

The training uses the Windows10 operating system. The server uses the NVIDIA GeForce GTX1650 graphics card, and the memory is 6 GB. Mask-RCNN-inception_v2 is used as a pre-trained model. Two sets of training parameters are used for experimental comparison: (1) max training steps are 200,000, the initial learning rate is set as 0.0002, momentum optimizer value is 0.9; (2) max training steps are 300,000, the initial learning rate is 0.0001, momentum optimizer value is 0.9. Using the first training parameters, the network will recognize the object, which is not Gimbap. In addition, the network will repeatedly identify the same object as different classes. Therefore, this article will choose the second set of parameters. Training takes about 27.7 h to run. On the training set, the total loss, classification loss, localization loss, and mask loss of the network are changed along with increasing training steps, as shown in Figure 8. After 300,000 steps, the total loss curve value of the training set is close to and falls below 0.2, oscillating in a small range. The network produces the correct result output for most image samples.

Figure 9 shows Gimbap's sample original images and instance segmentation images. The column (a) images are original images, which include Gimbap1 class and Gimbap2 class, as discussed in Section 3.1. Input the original image into the trained Mask R-CNN model, and the results are as column (b) images. Target Gimbaps are covered by different colors of masks, in the pictures. Each Gimbap is in a separated bounding box. The content in the upper left corner of the bounding box is the predicted label and score of the target Gimbap in the bounding box. Target Gimbap classification in the bounding box is Gimbap1 and Gimbap2. The probability that the target Gimbap belongs to this class is 100%.

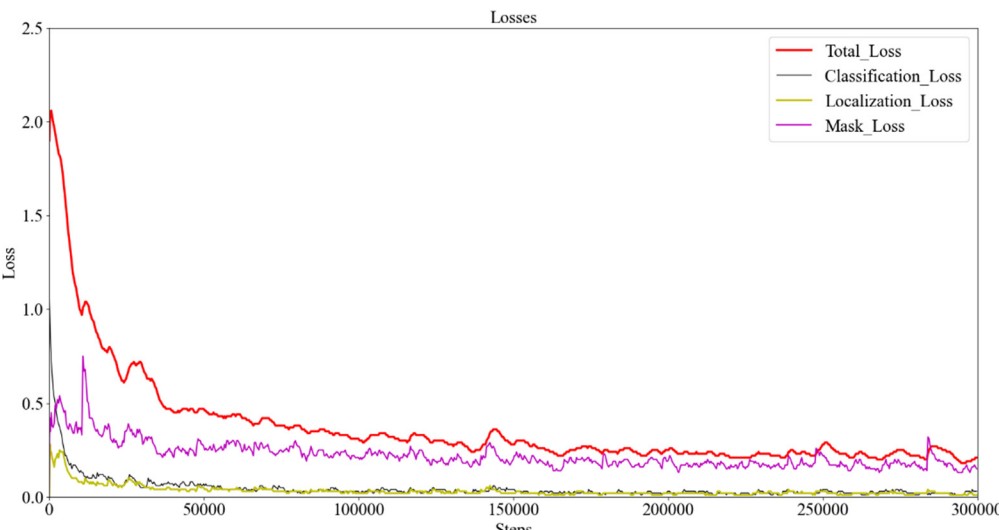

**Figure 8.** Relationship between loss and steps.

The Gimbap is detected by the Mask R-CNN model. The effect of the Mask R-CNN Gimbap detection model is evaluated by the Intersection over Union (IoU) between the predicted Gimbap bounding box and ground truth bounding box. The IoU is calculated as:

$$\text{IoU} = \frac{Area\ of\ Intersection}{Area\ of\ Union} = \frac{area(B_p \cap B_{gt})}{area(B_p \cup B_{gt})} \tag{12}$$

where $B_p$ is the predicted bounding box and $B_{gt}$ is the ground truth bounding box. If IoU $\geq 0.5$, the detection result is considered a True Positive (TP); otherwise, if $0 < \text{IoU} < 0.5$, the result is considered a False Positive (FP). If a mask is generated without Gimbap in the image, the result is also considered a False Positive. If the image contains Gimbap, but it is not detected, the result is considered a False Negative (FN). If the image does not contain Gimbap and no mask was generated, the result is considered a True Negative (TN). As in (13), the precision indicates the ratio of the correct number of Gimbap identified by the model to the total number. In (14), Recall is the ratio of the number of Gimbaps correctly identified by the model to the number of Gimbaps that actually exist.

$$\text{Precision} = \frac{TP}{TP + FP} \tag{13}$$

$$\text{Recall} = \frac{TP}{TP + FN} \tag{14}$$

The Average Precision (AP) is the area under the precision-recall curve for detection. Based on [31], AP is calculated from the mean precision at a set of eleven equally spaced recall levels [0, 0.1, . . . , 1]:

$$\text{AP} = \frac{1}{11} \sum\nolimits_{r \in \{0,0.1,...,1\}} P_{interp}(r) \tag{15}$$

The precision for each recall level $r$ is interpolated by taking the maximum precision value for any recall value greater than $r$.

$$P_{interp}(r) = \max_{\tilde{r}:\tilde{r} \geq r} p(\tilde{r}) \tag{16}$$

where $\text{p}(\tilde{r})$ is the measured precision at recall $\tilde{r}$. After calculating the AP for each class, the mean average precision (mAP) is obtained. In this experiment, AP at 0.5 IoU for Gimbap1 is 88.13%, and AP at 0.5 IoU for Gimbap2 is 82.72%. mAP at 0.5 IoU is 85.43%.

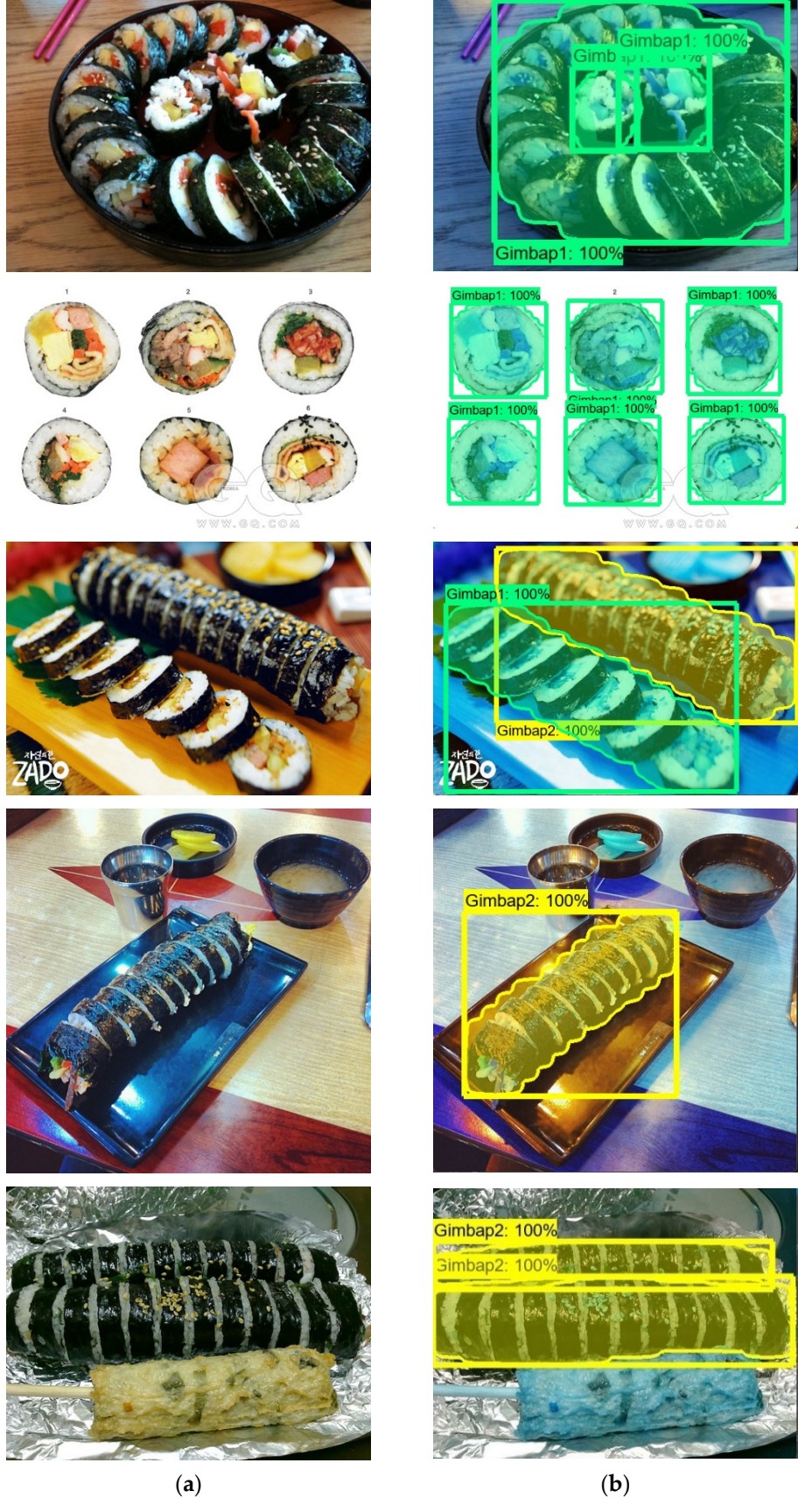

**Figure 9.** Experimental results of test samples: (**a**) original images: they include Gimbap1 class and Gimbap2 class, (**b**) instance segmentation images: Target Gimbaps are covered by different colors of masks.

### 5.2. Gimbap Calorie Estimation

Normally, the calories of one roll of Gimbap are 350–400 kcal. As discussed in Section 4, after resizing the image and calculating the physical size of each pixel, based on Gimbap detection results, calorie estimation results are shown in Figure 10. Figure 10a shows original images. In Figure 10(b1), the calories of each slice of Gimbap are in the range 35–40 kcal. For Gimbap2 class, there are two normal pieces of Gimbap and one big size slice of Gimbap, the total calories are 130.01 kcal. In Figure 10(b2), five pieces of Gimbap are recognized together as Gimbap1, and the calories are 194.57 kcal. The calories of the other two pieces of Gimbap are 35.07 kcal and 40.35 kcal, separately. In Figure 10(b3), the calories of the Gimbap are 390.84 kcal. There are some uncertainties, for example, the detection errors from the Mask R-CNN process and the calibration error from the calibration process, due to lighting conditions and image quality. However, the calorie estimation result is in the range of definition in real life. Therefore, the proposed Gimbap calorie estimation approach is effective.

### 5.3. Comparison with the Other Algorithm

The performance of the calorie estimation method with Mask R-CNN is evaluated by comparing its results with the results estimated by Faster R-CNN, as shown in Figure 11. The same test image is used for comparison, in order to minimize the effect of the parametric uncertainty. Five pieces of Gimbap are recognized as Gimbap1, and the calories are 113.3 kcal. Normally, the calories of five pieces of Gimbap are in the range of 175 kcal to 200 kcal. The calorie estimation result is much less than normal. After object detection using Faster R-CNN, only bounding box information is obtained, but no mask information. The unseen food shape cannot be considered. For the Gimbap1 label, based on Figure 7, there are two cases: (1) Gimbap is cut into pieces. The mask of the Gimbap is round. (2) Gimbap is cut into pieces and placed obliquely. The mask of the Gimbap is a rectangle. Since there is no mask information, it is hard to choose the right case. Therefore, the estimation result is wrong. In Figure 10(b2), the calorie estimation result is 194.57 kcal. It considered the unseen food and uses Mask R-CNN instance segmentation results to estimate the food's calories. Therefore, the result satisfied the standard.

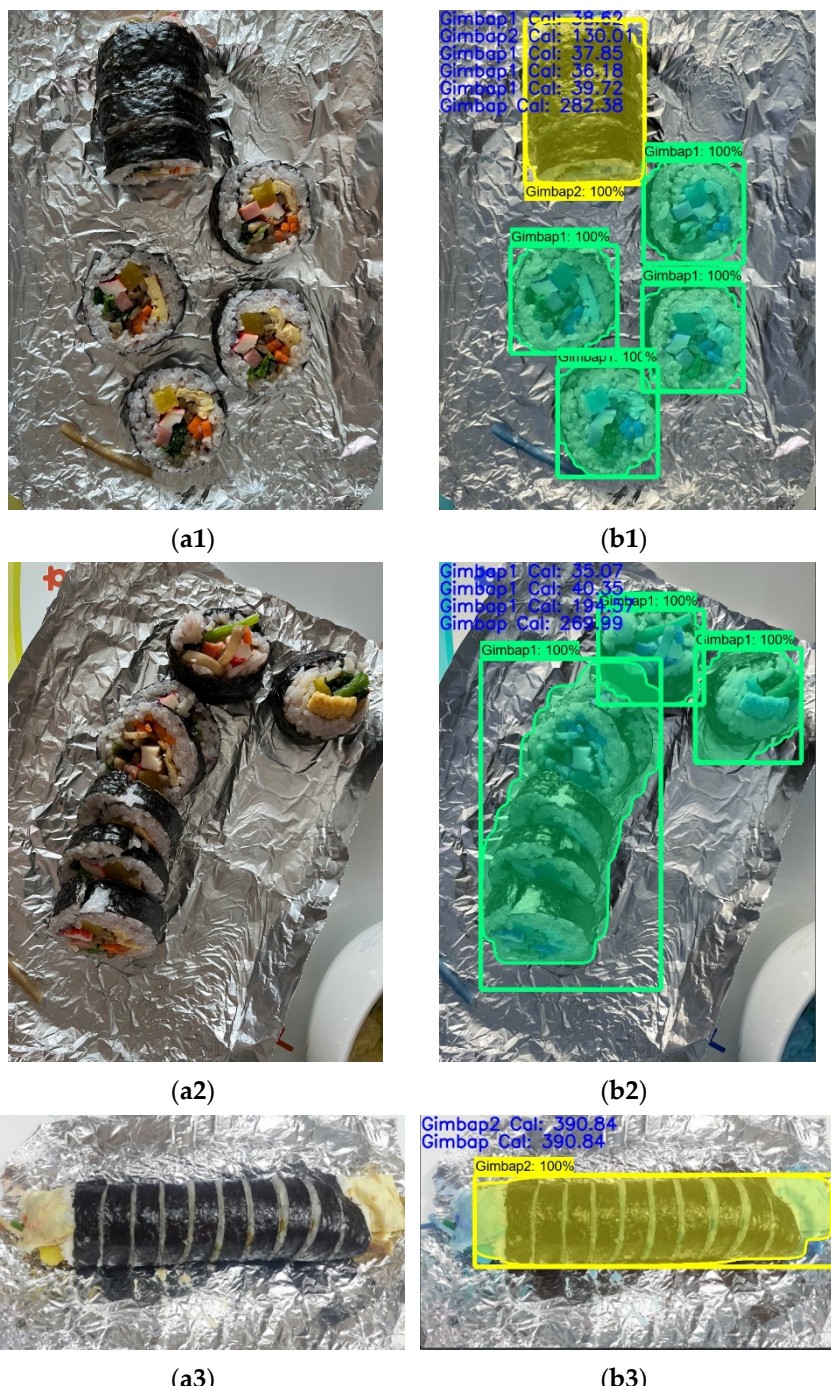

**Figure 10.** Calorie estimation results of test samples: (**a1**–**a3**) original images including Gimbap, (**b**) instance segmentation images. (**b1**) The Gimbap calorie of each detection is shown in the picture. The total calorie is 282.38 kcal. (**b2**), five pieces of Gimbap are recognized together as Gimbap1, and the calorie is 194.57 kcal. The calorie of the other two pieces of Gimbap is 35.07 kcal and 40.35 kcal, separately. The total calorie is 269.99 kcal. (**b3**), the calorie of the Gimbap is 390.84 kcal.

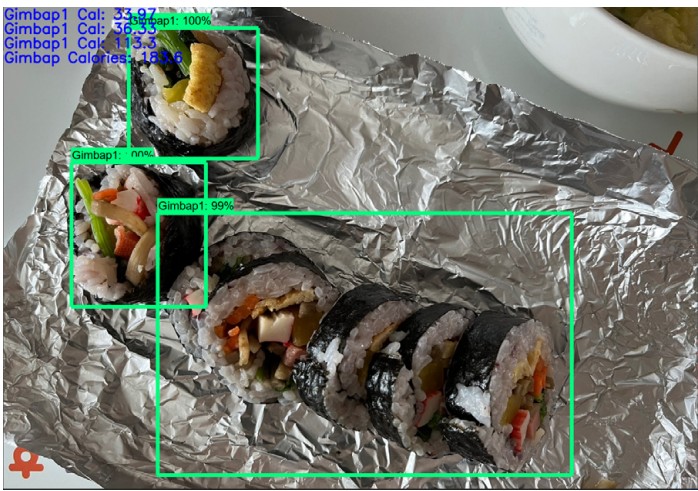

**Figure 11.** Calorie estimation result based on Faster R-CNN algorithm. Because it does not consider the unseen Gimbap, the calorie estimation result is much less than normal.

## 6. Conclusions

Recently, classifying food, estimating food quantity and nutritional content, and recording dietary intake have become important issues. In order to obtain accurate results, Mask R-CNN is applied for food instance segmentation and calorie estimation. Gimbap is chosen as an example of solid-volume food to show the effectiveness of the proposed methods. Firstly, based on the posture of Gimbap in plates, the labeled approach for the Gimbap image datasets is described, in order to process the Gimbap food segmentation and calorie estimation system. Secondly, an optimized model is created by fine-tuning Mask R-CNN architecture to obtain classification, bounding box, and mask information. In order to apply the method in real life, distinguishing from other methods using 3D LiDAR data or RGB-D images data, this work uses RGB images to train the model and uses a simple monocular camera to test the result. After training, the model shows 88.13% for Gimbap1 in AP (0.5 IoU) and 82.72% for Gimbap2 in AP (0.5 IoU). mAP at 0.5 IoU reaches 85.43%. Thirdly, a novel calorie estimation approach is proposed, combining the calibration result and the instance segmentation result. the calorie estimation approach can be extended to any solid-volume food, such as pizza, cake, burger, fried shrimp, oranges, and donuts. Finally, the effectiveness of the proposed approaches is proved. The Gimbap can be detected well. Based on Gimbap's segmentation instance result, Gimbap's calorie is correctly estimated. The calorie estimation result is close to Gimbap's calorie definition. Compared to other vision-based methods, the method in this paper outperforms in the accuracy of food segmentation and calorie estimation, since it uses the mask information from the Mask R-CNN algorithm and considers unseen food.

**Author Contributions:** Conceptualization, Y.D. and K.L.; methodology, Y.D.; software, Y.D.; validation, Y.D.; formal analysis, Y.D.; investigation, Y.D.; resources, Y.D.; data curation, Y.D.; writing—original draft preparation, Y.D.; writing—review and editing, Y.D. and K.L.; visualization, Y.D.; supervision, K.L.; project administration, S.P.; funding acquisition, K.L. All authors have read and agreed to the published version of the manuscript.

**Funding:** This work was supported by Korea Institute for Advancement of Technology (KIAT) grant funded by the Korea Government (MOTID)(P0008473, HRD Program for Industrial Innovation).

**Institutional Review Board Statement:** Not applicable.

**Informed Consent Statement:** Not applicable.

**Data Availability Statement:** Not applicable.

**Conflicts of Interest:** The authors declare no conflict of interest.

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
