# Peer review of "Utilizing Mask R-CNN for Solid-Volume Food Instance Segmentation and Calorie Estimation"

_applsci, doi:10.3390/app122110938_

Round 1
Reviewer 1 Report
The paper applies Mask R-CNN for food instance segmentation. However, considering that there exist many instance segmentation methods, it is essential to compare them in terms of both accuracy and runtime efficiency.
The statistics of the dataset should be provided.
I am wondering whether it is possible to predict the estimation of calories directly from MaskRCNN, e.g., by introducing a new branch.
The related work is not comprehensive. Some relevant instance segmentation works should be included, e.g., Differentiable multi-granularity human representation learning for instance-aware human semantic parsing.
Reviewer 2 Report
The manuscript entitled “Utilizing Mask R-CNN for Solid Volume Food Instance Segmentation and Calorie Estimation” has been investigated in detail. The topic addressed in the manuscript is potentially interesting and the manuscript contains some practical meanings, however, there are some issues which should be addressed by the authors:
1) In the first place, I would encourage the authors to extend the abstract more with the key results. As it is, the abstract is a little thin and does not quite convey the interesting results that follow in the main paper. The "Abstract" section can be made much more impressive by highlighting your contributions. The contribution of the study should be explained simply and clearly.
2) The readability and presentation of the study should be further improved. The paper suffers from language problems.
3) The “Introduction” section needs a major revision in terms of providing more accurate and informative literature review and the pros and cons of the available approaches and how the proposed method is different comparatively. Also, the motivation and contribution should be stated more clearly.
4) The importance of the design carried out in this manuscript can be explained better than other important studies published in this field. I recommend the authors to review other recently developed works.
5) What makes the proposed method suitable for this unique task? What new development to the proposed method have the authors added (compared to the existing approaches)? These points should be clarified.
6) “Discussion” section should be added in a more highlighting, argumentative way. The author should analysis the reason why the tested results is achieved.
7) The authors should clearly emphasize the contribution of the study. Please note that the up-to-date of references will contribute to the up-to-date of your manuscript. The studies named "Detection of solder paste defects with an optimization‐based deep learning model using image processing techniques; Optimization of deep learning model parameters in classification of solder paste defects "- can be used to explain the method in the study or to indicate the contribution in the “Introduction” section.
8) The effect of the parametric uncertainty is not discussed in detail. How did the comparison methods perform with or without the uncertainty?
9) It will be helpful to the readers if some discussions about insight of the main results are added as Remarks.
This study may be proposed for publication if it is addressed in the specified problems.
Author Response
Please see the attachment, thank you.

Reviewer 3 Report
The article titled “Utilizing Mask R-CNN for Solid Volume Food Instance Segmentation and Calorie Estimation” uses R-CNN for real-time vision-based method for solid volume food instance segmentation and calorie estimation is utilized and they train RGB images for testing and evaluation of the proposed technique.
The article is relevant, but the overall aspects are lacking as follows.
1. Authors need to seriously improve on the overall formatting of this paper.
2. The font and size of all equations need to be revised.
3. The formatting of pseudo-code is very poor.
4. Authors are suggested to enlist their contributions at the end of the Introduction section.
5. In the pseudo-code for the proposed technique, what do foods with random combinations represent according to the defined problem statement?
6. How do classes of calories are computed? Provide a proper equation or computation method for it.
7. Why authors used 300000 max training steps?
8. Authors need to add the epochs vs learning rate comparison for their technique. Because the training is consuming a lot of execution time.
Author Response
Please see the attachment, thank you.

Round 2
Reviewer 1 Report
The revision has addressed my concerns.
Reviewer 2 Report
All my comments have been thoroughly addressed. It is acceptable in the present form.